# Relaxation of bosons in one dimension and the onset of dimensional crossover

Chen Li[1,2], Tianwei Zhou[1,3], Igor Mazets[2,4], Hans-Peter Stimming[4],
Frederik S. Møller[2], Zijie Zhu[5], Yueyang Zhai[6], Wei Xiong[1],
Xiaoji Zhou[1], Xuzong Chen[1*] and Jörg Schmiedmayer[2†]

**1** School of Electronics Engineering and Computer Science,
Peking University, Beijing 100871, China
**2** Vienna Center for Quantum Science and Technology (VCQ),
Atominstitut, TU-Wien, Vienna, Austria
**3** INO-CNR Istituto Nazionale di Ottica del CNR,
Sezione di Sesto Fiorentino, I-50019 Sesto Fiorentino, Italy
**4** Research Platform MMM "Mathematics–Magnetism–Materials",
c/o Fakultät für Mathematik, Universität Wien, 1090 Vienna, Austria
**5** Institute for Quantum Electronics, ETH Zurich, 8093 Zurich, Switzerland
**6** Science and Technology on Inertial Laboratory, Beihang University, Beijing 100191, China

* xuzongchen@pku.edu.cn, † schmiedmayer@atomchip.org

## Abstract

We study ultra-cold bosons out of equilibrium in a one-dimensional (1D) setting and probe the breaking of integrability and the resulting relaxation at the onset of the crossover from one to three dimensions. In a quantum Newton's cradle type experiment, we excite the atoms to oscillate and collide in an array of 1D tubes and observe the evolution for up to 4.8 seconds (400 oscillations) with minimal heating and loss. By investigating the dynamics of the longitudinal momentum distribution function and the transverse excitation, we observe and quantify a two-stage relaxation process. In the initial stage single-body dephasing reduces the 1D densities, thus rapidly drives the 1D gas out of the quantum degenerate regime. The momentum distribution function asymptotically approaches the distribution of quasimomenta (rapidities), which are conserved in an integrable system. In the subsequent long time evolution, the 1D gas slowly relaxes towards thermal equilibrium through the collisions with transversely excited atoms. Moreover, we tune the dynamics in the dimensional crossover by initializing the evolution with different imprinted longitudinal momenta (energies). The dynamical evolution towards the relaxed state is quantitatively described by a semiclassical molecular dynamics simulation.

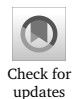
# 1 Introduction

The study of relaxation, thermalization and equilibration in an isolated many-body quantum system [1–4] has a long history starting with von Neumann [5]. An integrable system will not fully thermalize [6,7], but dephase towards a generalized Gibbs ensemble (GGE) [8–10], reflecting its many conserved quantities. Bosons in one dimension (1D) [11,12] are a model system to study these fundamental questions at the interface between microscopic quantum evolution and statistical physics. If the interactions are short range, the physics is described by the integrable Lieb-Liniger model [13–15].

However, in physical realization in the laboratory, "integrability" is never perfectly maintained. If the system is only approximately integrable, the dephased state is pre-thermal and is expected to reach thermal equilibrium at a much later time [16,17]. The integrability of identical bosons in 1D with short-range interactions can be broken by numerous effects [18–27]. Moreover, in a real experimental implementation, the strict 1D condition is maintained only within a limited time frame. For example, inevitable imperfections and noise lead to heating, which drives the system into the dimensional crossover regime, breaking integrability to some extent [28,29].

The pioneering work of Kinoshita et al. [6] introduced the Newton's cradle into the microscopic world as a typical near-integrable model. Since then it has attracted broad interests in the quantum gas community [24–27]. In this "*quantum Newton's cradle*", opposite longitudinal momenta are imprinted on the atomic ensembles, which then oscillate in the 1D-trap and collide. If the imprinted momenta are much smaller than required to excite transverse excitations, the oscillations persist for many atomic collisions. In Kinoshita et al. [6] the 1D condition is guaranteed by applying a very shallow longitudinal trap, so that high energy particles can leave the 1D traps at both ends. In return, the system exhibits significant loss [30–32].

In our work, we revisit the quantum Newton's cradle setting and study the dynamics of the longitudinal momentum distribution function (MDF) and the transverse excitation at the onset of 1D-3D crossover [18,33,34]. Opposite to Ref. [6], we retain nearly all the atoms during the dynamical evolution, thus significantly extend the observation time up to 400 oscillation periods. The non-equilibrium dynamics occur in two stages, dominated by different mechanisms. At a short time scale, the single-body dephasing reduces the 1D density, thus rapidly drives the 1D system out of the degenerate regime and asymptotically approaches the non-degenerate limit. During this time, the oscillation period averaged distribution of quasimomenta (also referred to as rapidities) is nearly conserved, while the MDF deforms significantly and finally in the non-degenerate limit tends to coincide with the quasimomentum distribution. At longer times, the system smoothly evolves into a dimensional crossover from 1D to 3D with a small fraction of the atoms transversely excited. The system finally relaxes to an equilibrium with a Gaussian momentum distribution suggesting a thermal final state.

Within the non-degenerate regime, accurate predictions for the system can be obtained very efficiently using a simple molecular dynamics (MD) calculation [35–38]. MD is a well-established numerical tool of simulation of many-body dynamics in various fields such as physics of fluids or chemical physics. In MD, the center-of-mass dynamics of particles (molecules) are classical, subject to the Newtonian mechanics. The internal degrees of freedom can be discretized (quantized). The main prerequisite to use the MD is therefore the non-degeneracy of the system, which is necessary to allow for the classical description of center-of-mass trajectories of individual molecules in a simulation. Alternatively, one could describe the system using the recently developed theory of generalized hydrodynamics (GHD), which has been proposed to describe dynamics of 1D quantum gases at or close to the integrable point [27,39–43]. Unlike other methods, GHD takes into account interactions between particles and remains across all phases of the Lieb-Liniger model. Recently, we have extended the applicability of GHD to the dimensional crossover regime by including transversely excited states [44]. Although that approach displayed very promising results on describing realistic systems in short to intermediate time scales, employing the full machinery of GHD to describe a primarily non-degenerate gas seems disproportionate. Therefore, within the present paper's scope, we opt for the much simpler MD simulations, whose efficiency allows us to account for the whole spectrum of transverse excitations in an easy and efficient way and makes it more suitable for describing the intermediate to long-term dynamics. The simulation results show an excellent quantitative agreement with the experimental observations.

The paper is organized as follows. In Sec. 2, we introduce the experimental setup, methods,

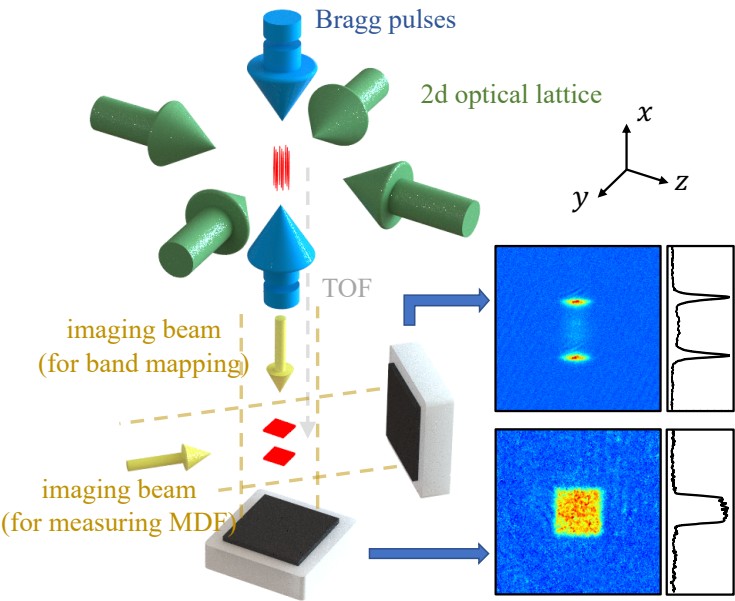

Figure 1: Experimental setup. The 1D Bose gas is prepared by adiabatically loading a BEC into a 2D optical lattice (green arrows). A sequence of Bragg pulses (blue arrows) is applied to excite the atoms to oscillate and collide in the 1D traps. After some duration, the atoms are released from the 1D traps and detected after TOF (yellow arrows), providing us the longitudinal MDF (horizontal imaging) and the fractional population of atoms in transverse states (vertical imaging).

and conditions. In Sec. 3, we present the experimental observations of MDF. In Sec. 4, we evaluate the dephasing process and observe that it is independent of the 1D density. In Sec. 5, we study the relaxation of MDF under the near-integrable condition, as the 1D systems evolve from the degenerate to non-degenerate regimes. In Sec. 6-7, we introduce the MD simulation and study the dynamical process towards thermalization at the onset of 1D-3D crossover. We compare the predictions of our simulation to experimental measurements in both longitudinal and transverse degrees of freedom. In Sec. 8, we further explore the dynamics initialized with a significant number of atoms having larger collision energies than the threshold of transverse excitation. In Sec. 9, we discuss the effect of virtual excitations as an alternative mechanism of breaking integrability in 1D. In Sec. 10, we draw the conclusions.

## 2 Experimental setup

### 2.1 Preparation and characterization of 1D gases

We start our experiment by preparing a $^{87}$Rb BEC in an all-optical trap. The BEC is then adiabatically loaded into a square array of 1D traps formed by a 2D optical lattice created with two retro-reflected perpendicular laser beams (see Fig. 1). The atoms are strongly confined in transverse directions ($y$ and $z$) in 1D traps with a trap frequency of $\omega_\perp/2\pi = 31.0(3)$ kHz and weakly confined in the longitudinal direction ($x$) with $\omega_\parallel/2\pi = 83.3(8)$ Hz (see Appendix A.1 for more details). The prepared 1D gases are characterized by the following **key parameters:**

**Atom number.** In the experiment, the atom number per tube varies across the 1D trap array. To balance the contributions from each tube, we calculate the weighted-average atom number.

The effect of the lattice loading procedure on the atom-number is considered by following the method proposed in Ref. [45]. The number of atoms per tube strongly depends on the total atom number, $N_{tot}$, of the BEC. By tuning $N_{tot}$ between $1 \times 10^4$ and $1 \times 10^5$, the atoms are distributed in 400 to 1200 tubes with a weighted-average atom number per tube, $N$, ranging from 40 to 130.

**Interaction strength.**   The interaction strength is characterized by the dimensionless Lieb-Liniger parameter $\gamma = c/n_{1D}$ (we use the standard notation for the Lieb-Liniger model: $c = mg_{1D}/\hbar^2$ and $n_{1D}$ is the mean 1D density) [12–14, 46]. To account for the inhomogeneous distribution of atoms over the tubes, we calculate the weighted average for $\gamma$. At the beginning of the evolution, $\gamma_0 = 1.5$ and $0.7$ for the systems with $N = 40$ and $130$, respectively. As the dephasing proceeds, the atoms spread along the tubes and the 1D density $n_{1D}$ drops, thus increasing $\gamma$. When the system is fully dephased, $\gamma \sim 6$ for $N = 40$, and $\gamma \sim 2.5$ for $N = 130$.

**Temperature and degeneracy.**   The temperature of the 1D gases is evaluated with the half-width at half-maximum (HWHM) of the Lorentzian-like longitudinal MDF before the Bragg pulses are employed. The inhomogeneous density profile is considered via the local-density approximation [47–50], and the significantly strong interaction is taken into account in a quantum Monte Carlo calculation [51]. We obtain the temperatures $T = 34\,\mathrm{nK}$ ($\widetilde{T} = 1.6$) and $T = 94\,\mathrm{nK}$ ($\widetilde{T} = 4.4$) for $N = 40$ and $N = 130$, respectively. The reduced temperature $\widetilde{T} = 2\hbar^2 k_B T/(m g_{1D}^2)$, together with $\gamma$, characterize the degeneracy of 1D gases. For $\widetilde{T} \gg 1$, the crossover from degenerate to non-degenerate regime occurs at $\gamma \simeq \widetilde{T}^{-1/2}$ [52]. As such, our Newton's cradle experiments start in the intermediate regime of degeneracy and approach the non-degenerate limit as the dephasing occurs.

**Heating rate.**   Heating is evaluated by holding a BEC in the identical trapping potential without a Bragg-pulse excitation. Within the time scale of our experiment ($4.8\,\mathrm{s}$), we observe excitation of about 3% of the atoms to the first transversely excited state and hardly any to the second state (see Appendix A.2).

**Tunneling between 1D tubes.**   Under our experimental conditions, the residual single-atom tunnel coupling ($J < 0.01\,\mathrm{s}^{-1}$) is too small to influence the longitudinal dynamics. The Josephson frequency in our system is $\omega_J = \sqrt{2J(2J + 2\mu/\hbar)} \ll \hbar k_L^2/m$, which holds even for higher transversely excited states $n_r = 1$ or $2$ which show an increased $J$. Here $\mu$ is the chemical potential. As such, we can safely neglect the effect of tunneling.

## 2.2   Excitation of longitudinal motion

To start the non-equilibrium dynamics in the longitudinal direction, a sequence of two Bragg pulses using retro-reflected $\lambda = 852\,\mathrm{nm}$ light is employed on the 1D gases (Fig. 1). Neutral atoms initially at rest are diffracted to a series of momentum states $e^{2ni\hbar k_{Bragg}x}|\psi_0\rangle$ when exposed to the standing wave light pulses, where $k_{Bragg} = 2\pi/852\,\mathrm{nm}$ is the recoil momentum and $|\psi_0\rangle$ is the atom's (low-momentum) state before scattering [53–58]. Following the method described in Ref. [59] we can accurately control the resulting momentum distribution function (MDF) by tuning the pulse intensity $V_0$ and the time sequence of pulse lengths $t_1$, $t_3$, and separation $t_2$ (illustrated in Fig. 2). In this way we prepare a large variety of initial MDFs from where our study starts. The ones used in this paper are presented in Fig. 3 and provide us with varying fractions of atoms with momenta $|k| > k^{th}$. Here $k^{th} = \sqrt{2m\omega_\perp/\hbar}$

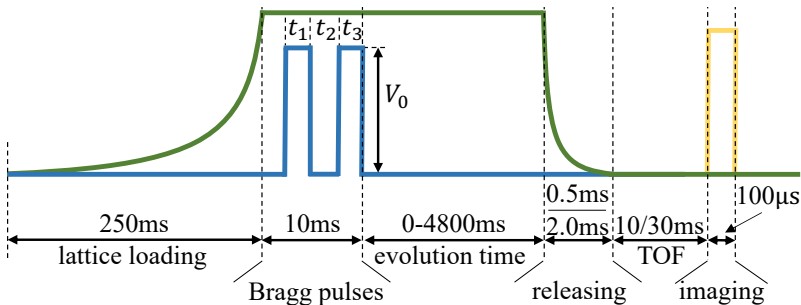

Figure 2: Experimental sequence (not to scale). Green: lattice depth as a function of time. The lattice is turned off in 0.5 ms for measuring the longitudinal MDF and in 2 ms for measuring the transverse excitation. Blue: Bragg pulses characterized by the pulse intensity $V_0$ and the time sequence $[t_1, t_2, t_3]$. Yellow: imaging pulse applied after a TOF of 10 ms (for $N = 40$) or 30 ms (for $N = 130$).

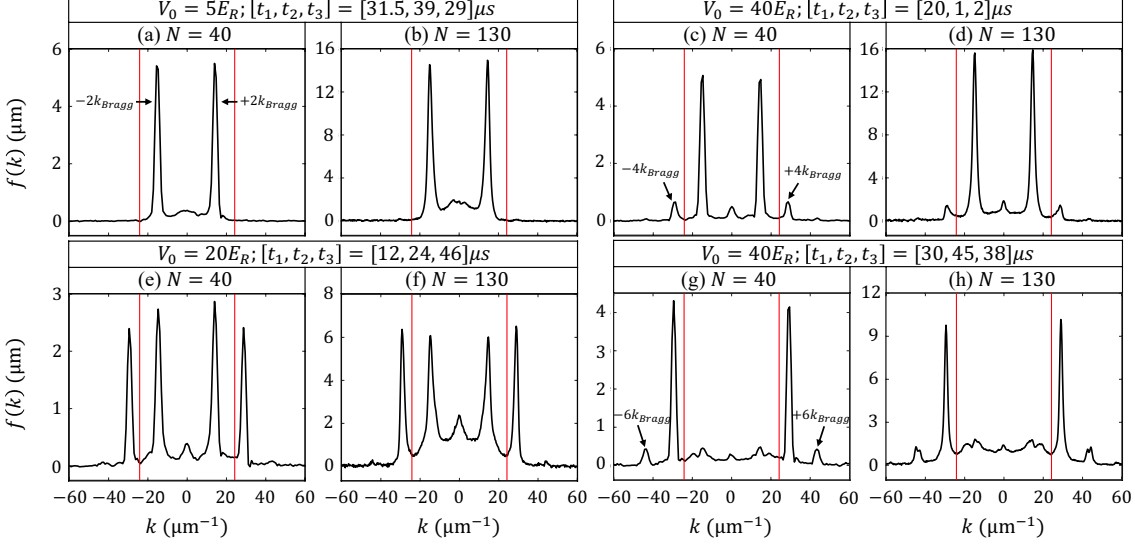

Figure 3: Initial MDFs (experimental measurements) and their corresponding Bragg pulse sequences. The pulse intensity $V_0$ is expressed in units of $E_R = (\hbar k_{Bragg})^2/2m$. The red vertical lines indicate the momenta $\pm k^{\text{th}}$.

is the threshold momentum, and two head-on colliding atoms with momenta $\pm k^{\text{th}}$ have the required collision energy ($E^{\text{th}} = 2\hbar\omega_\perp$) for populating the transversely excited states [18].

The momenta $\pm 2k_{Bragg}$ correspond to $\sim 40\%$ of the excitation energy $E^{\text{th}}$, while $\pm 4k_{Bragg}$ and $\pm 6k_{Bragg}$ are beyond the excitation threshold. In Sec. 3-7, we will first investigate the dynamics initialized with the distributions shown in Fig. 3 (a) and (b), where hardly any atom has the energy to be transversely excited at the beginning of dynamical evolution[1]. In Sec. 8, we will further show the results starting with higher-energy distributions shown in Fig. 3 (c)-(h).

---

[1]The quantities that characterize the dynamics during a collision process are actually the quasimomenta rather than the bosonic momenta of two particles. Owing to the interaction, the quasimomentum distribution is in general broader than the MDF. Thus, we expect $< 0.1\%$ and $\sim 2\%$ of atoms obtain the quasimomenta beyond the excitation threshold for $N = 40$ and $N = 130$, respectively. These values are estimated according to the method described in Sec. 6.

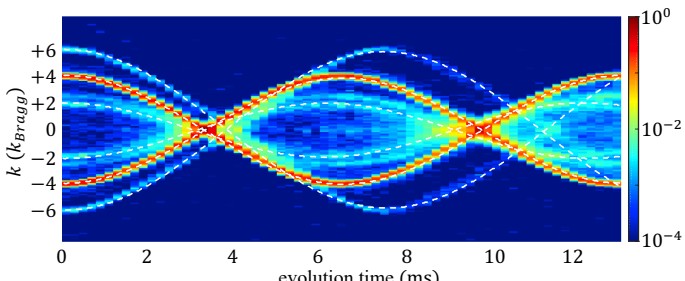

Figure 4: Oscillations in the first period (time resolution of measurements $\Delta t = 0.2\,\text{ms}$). The distributions are shown in log scale with the maximum normalized to unity. The oscillation periods for $\pm 2k_{Bragg}$, $\pm 4k_{Bragg}$, $\pm 6k_{Bragg}$ are measured to be $12\,\text{ms}$, $13.2\,\text{ms}$, $15.2\,\text{ms}$, respectively.

## 2.3 Post-pulse evolution and detection

After the Bragg pulses, we keep the lattice on for some duration $t$, during which the atoms oscillate in the tubes with a period of $\mathcal{T} = 12\,\text{ms}$ and collide with each other. At the end of evolution, the atoms are released from the optical lattice and expand in 3D for a long time-of-flight (TOF). The image taken in the horizontal plane provides us a density profile given by the MDF in the longitudinal direction of 1D gases. Alternatively, we apply the imaging beam vertically and detect the population of atoms in transverse states (see Fig. 1). See Appendix A.3-A.4 for more technical details on the methods of detection and data analysis.

## 3 Basic observations: two-stage relaxation process

To study the non-equilibrium dynamics after the Bragg pulses, we explore the MDF $f(t,k)$ for an evolution time up to $4.8\,\text{s}$ (400 oscillation periods). The early stage of evolution is dominated by the oscillation of the kicked momentum components in the longitudinal trap, as shown in Fig. 4 for strong excitation pulses (Fig. 3 (g)). The momentum peaks $\pm 2k_{Bragg}$, $\pm 4k_{Bragg}$, and $\pm 6k_{Bragg}$ exhibit different oscillation periods, which stem from the anharmonicity of the longitudinal confinement in the tubes, caused by the Gaussian profile of the lattice beams. The measurements are consistent with the expectations based on our trap parameters within reasonable experimental imperfections, for example, the nonideal lattice beam quality, the imperfect overlap between lattice beams, etc.

Fig. 5 shows the dynamics of MDF for the case of excitation with only $\pm 2k_{Bragg}$ momentum components (Fig. 3 (a) and (b)). From these measurements, one clearly observe two distinct stages:

**Stage I, single-body dephasing:** With increasing time, the clear oscillations in the early stage become more and more blurry and finally completely dephase after about 60 oscillations (720 ms). Even though the MDFs behave markedly different, the dephasing seems to be happening over a similar time scale in both cases (atom numbers per tube: $N = 40$ and $N = 130$). We will quantitatively compare the dephasing processes in these two cases and discuss them in more detail in Sec. 4 (Stage Ia). Accompanied by the dephasing process, the Bragg peaks in the MDF broaden and become rounded over time at a near-integrable point. This deformation is also observed on the oscillation period averaged profile, and it is expected to be induced by the 1D density decrease caused by the dephasing effect. We will investigate this observation in Sec. 5 (Stage Ib).

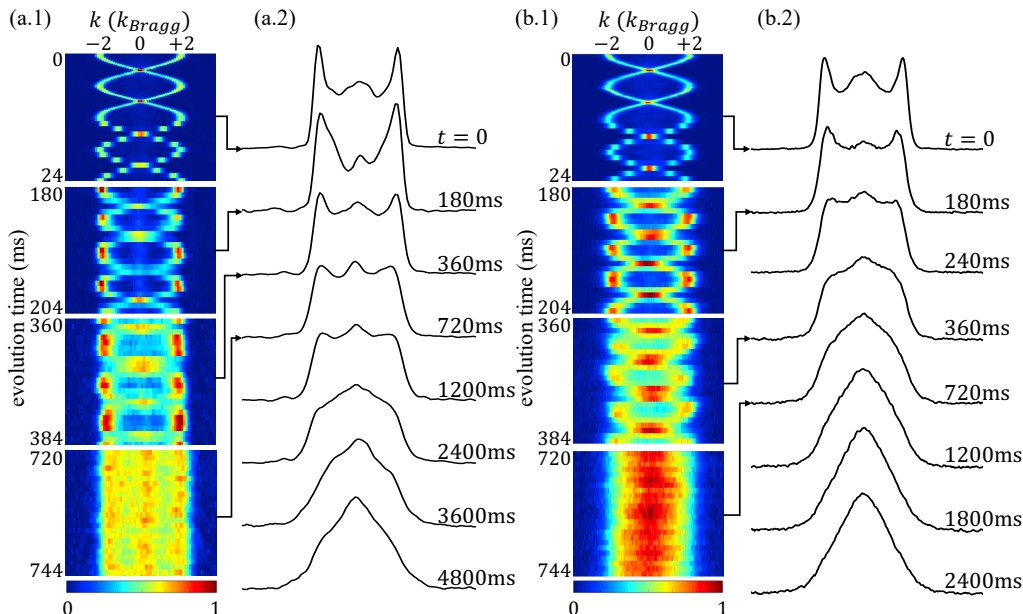

Figure 5: Basic observations: MDF for dynamics starting with $\pm 2k_{Bragg}$ (excitation Fig. 3 (a) and (b)). (a.1, b.1) Four segments of the dynamics of MDF starting from the 1st period (upper), the 15th, 30th to the 60th period (bottom) for $N = 40$ (a) and $N = 130$ (b). Each segment includes two oscillation periods. The time resolution of the measurements in the first period is finer than the following measurements by a factor of 5. One clearly observes a blurring of the oscillations which can be attributed to dephasing. (a.2, b.2) Time evolution of the oscillation period averaged MDF $F(t, k)$: for $N = 40$ (a) and $N = 130$ (b). For long times the MDF relaxes towards a Gaussian profile.

**Stage II, relaxation towards thermal equilibrium:** At long times ($t > 720$ ms, 60 periods), the MDF is completely dephased and does not show short time variations within one oscillation period. The MDF further evolves towards a Gaussian distribution. We conjecture that the observed long time relaxation is dominated by the scattering processes mainly between atoms in the transverse ground state and a small population in the transversely excited state. The two-body collisions involving transversely excited atoms allow redistribution of the longitudinal momenta, thus breaking the integrability and constituting the onset of the crossover between 1D and 3D. As we will show in Sec. 6, this conjecture is supported by the excellent agreement between the experimental observations and the results of MD simulations implemented on a semiclassical model using experimentally determined parameters as input.

## 4 Stage Ia: single-body dephasing

The simplest and most direct explanation of the observed blurring of the oscillations in Fig. 5 is the single-body dephasing. It does not require interactions and is caused by a diffusion of the relative oscillation phase of each particle. In our experimental setup, the single-body dephasing has two distinct contributions: (*i*) The anharmonicity of the longitudinal Gaussian confinement within each tube makes the atom with larger kinetic energy oscillate at a slightly larger period as can be observed in Fig. 4. This effect leads to the dephasing of atoms with different energies within each tube. (*ii*) The inhomogeneity among tubes, which is also a result of the Gaussian profile of the lattice beams, leads to different oscillation frequencies in

different tubes and to dephasing of the oscillations among the tubes.

Compared to the inhomogeneity of the traps, we are concerned more about the anharmonicity of longitudinal confinement, which broadens the longitudinal spatial distribution of atoms in each tube. The resulting decrease of atomic density drives the 1D gases out of the degenerate regime. We will demonstrate this effect in Sec. 5.

The dephasing takes effect until the MDF stops varying during one oscillation period. The process can be characterized by the deviation of the MDFs from their oscillation period averaged profile,

$$\mathcal{D}(t) = C_{\mathcal{D}} \int dk \int_{\mathcal{T}} dt' \left[ f(t+t', k) - F(t,k) \right]^2 , \tag{1}$$

where $F(t,k)$ is the average profile of MDF over one oscillation period. For the convenience of comparison, the prefactor $C_{\mathcal{D}} = 2 \times 10^7/N^2$ rescales the calculation according to the atom number and lifts the results close to 1.

In Fig. 6 we plot $\mathcal{D}(t)$ as a function of time for $N = 40$ and $N = 130$. $\mathcal{D}(t)$ decreases as the variation of MDF in one period decays over time. The dephasing we observe is independent of the atomic density. In both cases, the MDFs are fully dephased at around 720 ms (marked with the vertical dashed line), even though the dephased distributions are distinct due to the different relaxation rates (see Sec. 6). Beyond 720 ms $\mathcal{D}(t)$ reaches a plateau, which is dominated by the imaging noise floor. The density-independent dephasing rate can be explained by a collisionless classical calculation with our experimental conditions [25].

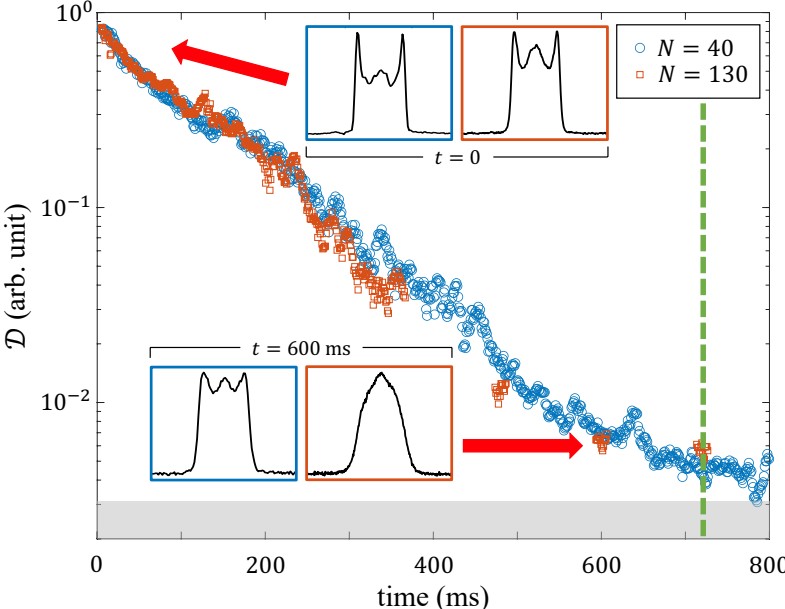

Figure 6: Single-body dephasing for $N = 40$ and $N = 130$. The dephasing rate is nearly independent of the atomic density. The insets show the oscillation period averaged MDFs at the indicated evolution times, and the boxes' colors indicate the corresponding data-sets. After about 720 ms (marked with the green dashed line), the MDF does not significantly vary over time anymore. The shaded area indicates the noise floor.

# 5 Stage Ib: dynamics of the momentum distribution function at a near-integrable point

To study the long time evolution, we average out the effect of single-body dephasing by calculating the oscillation period averaged MDF $F(t, k)$. As the relaxation occurs, $F(t, k)$ evolves from a double-peak profile to a peak-rounded profile and finally approaches a Gaussian distribution. The evolution is characterized by the deviation of $F(t, k)$ from its closest thermal distribution. More specifically, we calculate the summed square of residuals between $F(t, k)$ and its best fit to a Gaussian distribution $\hat{F}(t, k)$

$$\mathcal{R}(t) = C_{\mathcal{R}} \int \mathrm{d}k \left[ F(t, k) - \hat{F}(t, k) \right]^2,$$

(2)

where $C_{\mathcal{R}} = 10^4/N^2$. This approach has emerged as the most robust way to characterize the distance from a relaxed Gaussian (thermal) equilibrium state under our experimental conditions with inevitable noise and fluctuations[2].

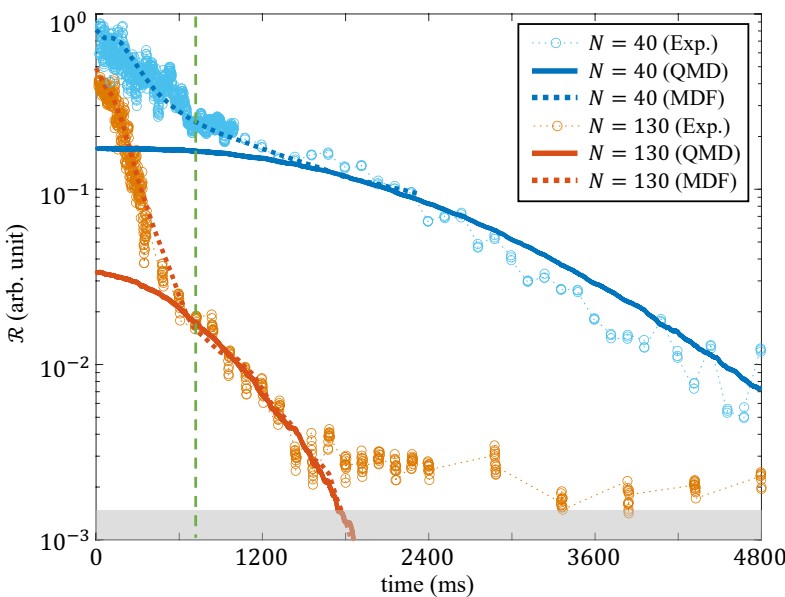

Figure 7: Relaxation characterized by the deviation of the oscillation period averaged MDF from its best-fit Gaussian distribution. The experimental measurements are shown with open circles in two colors for $N = 40$ and $N = 130$, respectively. These measurements are compared with the molecular dynamics simulation results, including both the calculated quasimomentum distributions (QMD) and the corresponding MDFs. In Stage I of evolution, the dynamics are well described by the evolution of the estimated MDF (dotted curves). As the system approaches the non-degenerate limit, the estimated MDF tends to coincide with the QMD (solid curves). In Stage II of evolution, the 1D gases relax towards a Gaussian MDF, which can be interpreted as a thermal equilibrium state. The molecular dynamics simulations agree nicely with the experimental data in Stage II. The vertical dashed line indicates the end of single-body dephasing and demarcates the two stages of relaxation. The shaded area indicates the noise floor.

In Fig. 7, $\mathcal{R}(t)$ is plotted with open circles for the two cases associated with different atom

---

[2]This method is a significant advantage over the conventional methods of Gaussian test, like the Kolmogorov-Smirnov test, Lilliefors test, and Kurtosis, which are all sensitive to high-momentum tails and noise.

numbers ($N = 40$ and $N = 130$). Two distinct stages characterized by different decay rates of $\mathcal{R}(t)$ emerge within the entire time scale, and the boundary between two stages presents at the time when the single-body dephasing ends (marked with the vertical dashed line). A similar phenomenon has been reported by Tang et al. in the dipolar Newton's cradle experiments [25]. We clarify that the rapid decay of $\mathcal{R}(t)$ in Stage I of evolution stems from the evolution of MDF within the integrable model. In the degenerate limit, the MDF of a 1D Bose gas is described by a Lorentzian-like profile and dominated by the phase fluctuations (see Appendix C or Ref. [49, 60]). The MDF is in general much narrower than the distribution of quasimomenta [27, 61–67], which are the conserved quantities characterizing the integrable many-body system. As a result of the single-body dephasing derived from the anharmonicity of confinement, the 1D densities decrease and asymptotically approach one-fourth to one-third of the initial values. Thus, at the end of Stage I, $\gamma \widetilde{T}^{1/2}$ is greater than 7.6 ($N = 40$) and 5.2 ($N = 130$) [3], suggesting that the dephased gases are in the non-degenerate regimes. The change of the interparticle interaction during this process deforms the MDF and brings it close to the quasimomentum distribution. As we observe in our experiments, the narrow momentum peaks present at the beginning of evolution are washed out and become rounded, resulting in the rapid decay of $\mathcal{R}(t)$ in the first 720 ms. When the single-body dephasing comes to its end, the decrease of density stops, and the evolution towards the non-degenerate limit is greatly slowed down. The process of the MDF approaching the quasimomentum distribution is illustrated in Fig. 8. Since the description of this figure involves the MD simulation that will be introduced in Sec. 6, we will explain the details later in the relevant paragraphs.

Note that the integrability is (nearly) preserved during Stage I, especially for the $N = 40$ case. The heating process affects the dynamics very little at short to intermediate time scales. Due to the parity conservation, an inelastic collision occurs when at least two atoms appear in the first transversely excited state (or at least one atom in the second excited state) in the same tube. When the atom number per tube is relatively low as the case in our experiments, the effect of heating on the longitudinal motion is postponed. We will demonstrate this statement in more depth in Sec. 6.

# 6 Stage II: relaxation towards a Gaussian (thermal) momentum distribution and molecular dynamics simulation

At longer times, the gases relax towards thermal equilibrium through inelastic collisions that involve the transversely excited modes. This process is triggered by the minimal heating effect and the tiny population of atoms with momenta $|k| > k^{\text{th}}$, marking the onset of dimensional crossover. As shown in Fig. 7, $\mathcal{R}(t)$ in Stage II decreases until it reaches a plateau, which is dominated by the imaging noise floor, indicating that the MDF of the 1D gas is indistinguishable from a Gaussian (thermal) distribution.

To further study the relaxation in the dimensional crossover, we implement a molecular dynamics (MD) simulation in a semiclassical framework. The dynamical evolution is described by quasiparticles characterized by their spatial coordinates, quasimomenta, and transverse modes' occupations. In the longitudinal direction, the quasiparticles are initialized according to the thermodynamic Bethe ansatz [68] but move as classical point-like objects. The transverse degrees of freedom are treated as discrete quantum levels, which is the new key (quantum) ingredient of our model. This model is valid in our Newton's cradle experiments because the quantum correlations are strongly suppressed in the first ten oscillation periods (5% of the evolution time), as the chemical potentials of the 1D gases turn from positive to negative

---

[3]Here we assume a constant temperature during Stage I. But in the real case, any increase in temperature resulting from relaxation and heating will make $\gamma \widetilde{T}^{1/2}$ even larger.

resulting from the decreases of 1D densities. Note that this time is too short to appreciably change the oscillation period averaged quasimomentum distribution due to the interplay between the effects of atomic scattering and the longitudinal trapping, which is known to break down the integrability and to induce relaxation [27] (see also [23]). As the dephasing in each tube happens, the 1D gases enter the non-degenerate regime, where $\gamma \gg \widetilde{T}^{-1/2}$ is fulfilled (see Sec. 2.1 and 5). In the non-degenerate regime, the filling of states is much smaller than unity, whereby the effects of quantum statistics become negligible. When close to the non-degenerate limit, quasiparticles become individual atoms, and quasimomenta can be interpreted as usual momenta.

Scattering and transitions between these transverse states are calculated. The collisions between quasiparticles follow the parity selection rules, and the lowest excitation energy is $E^{\text{th}}$. The transition matrix elements determining the transition probabilities are obtained from quantum-mechanical calculations similar to those of Ref. [46]. The transverse state of a quasiparticle is specified by a number $n$ of transverse excitation quanta, with $n = 0$ corresponding to the ground state of the transverse motion. We do not resolve the degenerate sublevels but invoke the statistical weight (i.e., degeneracy) $w_n = n + 1$ of the corresponding state of an isotropic 2D harmonic oscillator.

A harmonic longitudinal potential is assumed in order to make calculations simple and fast. We performed two tests to accept this assumption. (*i*) We tested an anharmonic potential $U_0 \tanh^2(x/\Delta x)$ that admits analytic integration of the equations of motion. The parameter $U_0$ was taken equal to the lattice depth and the typical length scale $\Delta x$ was chosen to provide the harmonic potential $\frac{1}{2}m\omega_\parallel^2 x^2$ for $|x| \ll \Delta x$. Using these parameters, the effect of the anharmonicity of the potential was found to be small. (*ii*) We initialized the simulation of dynamics in a harmonic trap with a fully dephased distribution, where the two Bragg components are indistinguishable and overlap during the whole oscillation period. Compared to the normal case starting with two distinct Bragg peaks, we observed a very small effect on the calculated quasimomentum distributions within our model. In other words, the anharmonicity is important during the initial relaxation stage only and we restricted ourselves to the harmonic model.

Each numerical realization corresponds to a single tube. The number of quasiparticles per tube $N$ as an input parameter is set to the weighted-average value measured in experiments. The initial distribution of quasiparticles is sampled according to a thermal distribution calculated by solving the Bethe-ansatz equations at the measured temperature in an experimentally defined harmonic trap. Afterwards, each quasiparticle obtains a boost of quasimomentum $-2k_{Bragg}$ or $+2k_{Bragg}$ with equal probability $(1 - \eta)/2$. As such, we kick the quasiparticles with Bragg momenta, leaving a fraction of $\eta$ of quasiparticles at the trap center. To match with the experimentally measured initial MDFs, $\eta$ is set to be 9% and 20% for $N = 40$ and $N = 130$, respectively. Subsequently, we propagate the quasiparticles over time according to the functions described in Appendix B.

The change of transverse states of quasiparticles due to heating in the optical lattice is included in the model. Its probability per unit time per quasiparticle is denoted by $\Gamma$. We assume $N\Gamma\bar{\tau} \ll 1$, where $\bar{\tau} = 2\pi/(\omega_\parallel N^2)$ is the typical time between two subsequent atomic collisions. Within our simulation, we set $\Gamma = 0.0375\,\text{s}^{-1}$ for both $N = 40$ and $N = 130$, which is determined by the heating rates measured in experiments (as shown in Appendix A.2). Whenever a collision occurs (i.e., when coordinates of two neighboring atoms coincide), we check both the possibility of the change of transverse states for the colliding pair of quasiparticles and the possibility of the change of transverse states for all the quasiparticles. We generate a pseudorandom number $\zeta'$ uniformly distributed between 0 and 1. If $\zeta' < \exp(-N\Gamma\tau)$, where $\tau$ is the time elapsed since the previous collision, then no state change occurs. Otherwise, we pseudorandomly select one of the $N$ quasiparticles and change its transverse excitation num-

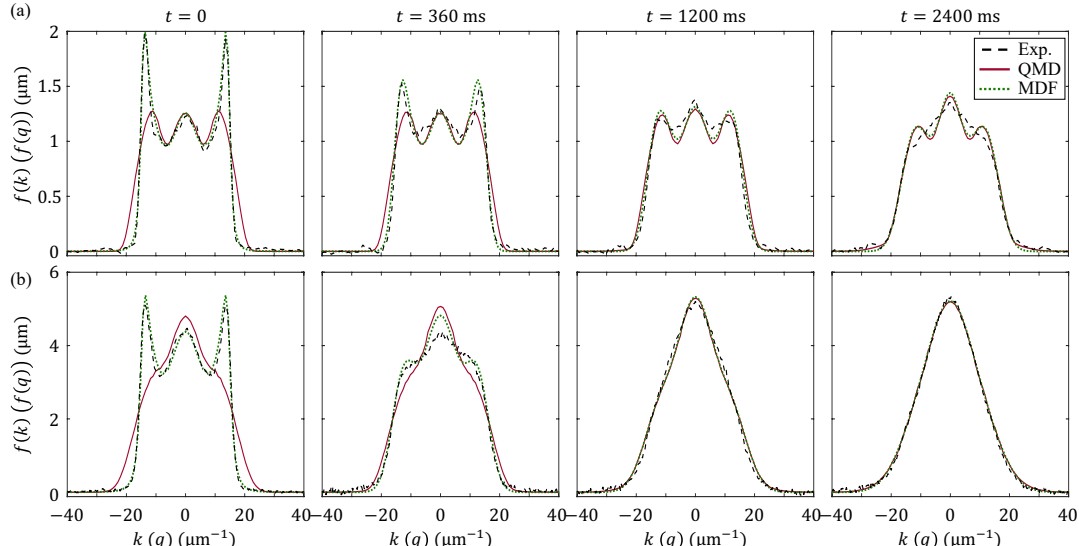

Figure 8: Estimation of oscillation period averaged MDF for (a) $N = 40$ and (b) $N = 130$. For comparison purposes, we estimate the MDFs (green dotted curves) from the quasimomentum distributions (QMD) calculated in MD simulations (red solid curves) and observe good agreement with the experimentally measured profiles (black dashed curves). As the 1D gases approach the non-degenerate limit, the MDF and QMD tend to coincide.

ber $n_j$ to $|n_j + 1|$ or $|n_j - 1|$, each channel having the probability of 50%. Since quasiparticles are predominantly in the ground state, the most probable process $n_j = 0 \rightarrow n'_j = 1$ leads to the energy supply to the system (heating).

The quasimomentum distribution derived from the MD simulation is distinct from the MDF (of the real physical bosons) when the system is far from the non-degenerate limit. For comparison with the experimentally measured profiles, we need to estimate the corresponding bosonic MDFs for the MD results. The relation between both distributions is not straightforward. There is no general analytic approach to calculate the MDF in the Lieb-Liniger model, and only numerical methods were for example applied in [27, 61–67]. Within the scope of this paper, we use an estimation of the MDF as outlined in Appendix C instead of an exact numerical calculation. Fig. 8 presents a comparison between the calculated quasimomentum distributions from the MD calculation (red), the estimated MDF (green) and the experimentally measured MDF (black dashed curves), which are all shown in oscillation period averaged profiles. The estimated MDFs are in good agreement with the experimental measurements throughout the entire dynamical evolution. Meanwhile, we observe the increasing similarity between the MDF and the quasimomentum distribution during Stage I of evolution.

The relaxation processes of the MD results are characterized by following Eq. (2), in the same way as in processing the experimentally measured profiles (see Fig. 7). $\mathcal{R}(t)$ calculated from both MDFs ($\mathcal{R}^{\mathrm{MDF}}$, dotted curves) and quasimomentum distributions ($\mathcal{R}^{\mathrm{QMD}}$, solid curves) are compared to the experimental results ($\mathcal{R}^{\mathrm{EXP}}$, open circles). In Stage I of evolution, $\mathcal{R}^{\mathrm{MDF}}$ displays similar features with $\mathcal{R}^{\mathrm{EXP}}$. The relaxation of MDF due to the transition from the degenerate to non-degenerate regimes is captured by the estimation of MDF. After the dephasing ends, $\mathcal{R}^{\mathrm{MDF}}$ asymptotically approaches $\mathcal{R}^{\mathrm{QMD}}$, and at longer times the two results tend to coincide.

Furthermore, it is demonstrated by the plot of $\mathcal{R}^{\mathrm{QMD}}$ that the system relaxes towards a Gaussian (thermal) distribution at an accelerating rate. As has been mentioned at the end of Sec. 5, the 1D condition of the cradle system is preserved until the transversely excited

states are populated with a threshold of atom number. For a 1D system with $N = 40$, it at least needs 5% of atoms in the first transversely excited state to break the integrability with inelastic collisions. These atoms may come from the minimal residual heating. From the plot of $\mathcal{R}^{QMD}$ for $N = 40$, we observe that it takes about 1 s to start the relaxation towards a Gaussian (thermal) momentum distribution. The onset of relaxation leads to an energy transfer from the longitudinal to transverse degrees of freedom (an increase of the population in the transversely excited states), which in return intensifies the relaxation itself.

In contrast to $N = 40$, the relaxation towards a Gaussian (thermal) momentum distribution occurs earlier and faster for $N = 130$. It is because that the higher atom number offers a larger chance for the atoms to be excited transversely, so that it equivalently lower the threshold. Additionally, owing to the higher collision rate, the excited atoms spend less time in the upper state before returning back to the ground state through collisions. During this process, the excitation energy is deposited to the ground state. While, in the lower atom number case, the excited atoms stay longer in the upper state, having a small but non-zero probability of being de-excited through external disturbances. Moreover, in the initial distribution of quasiparticles for the MD simulation, we expect about 2% of quasiparticles obtaining quasimomenta larger than $k^{th}$ for $N = 130$. This fraction of atoms is not observed in Fig. 3(b) because of the narrower profile of MDF compared to the quasimomentum distribution. In contrast, this fraction is expected to be almost zero for $N = 40$. The very rare high-energy quasiparticles speed up the relaxation to some extent. For the above reasons, we observe faster relaxation for $N = 130$.

## 7 Dynamics of transverse excitation

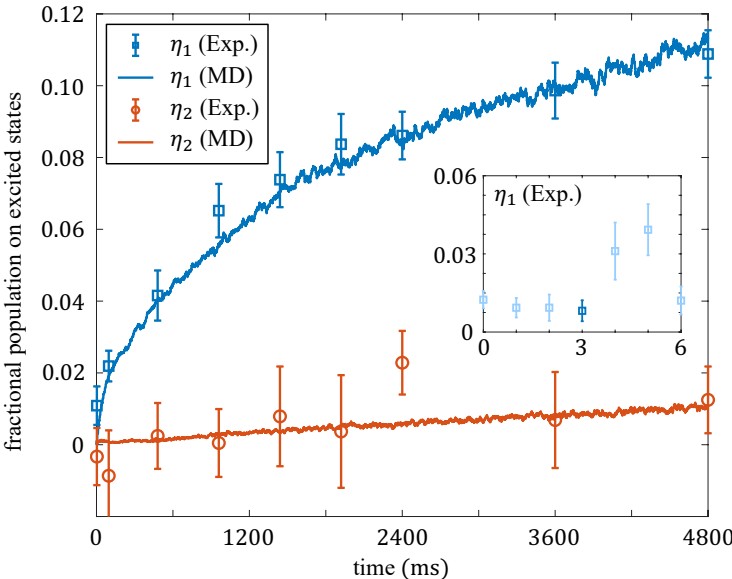

Figure 9: Fractional populations of atoms in the first (open squares) and second (open circles) transversely excited states versus evolution time ($N = 130$). The error bars denote the standard deviation of five measurements. The solid curves with corresponding colors show the results of the molecular dynamics (MD) simulations. The insert shows the measurements in the first 6 ms after the Bragg pulses, where the darker blue data point (measured at $t = 3$ ms) reflects the most credible in-trap excitation.

To further illuminate the relaxation process driven by the inelastic collisions, we study the dynamics of atoms in the transverse states by band mapping in the deep-lattice limit [69]. The fractional populations of atoms in the first and second transversely excited states (notated by $\eta_1$ and $\eta_2$, respectively) are extracted throughout the evolution from the experimental band mapping images (see Appendix A.4). We observe that approximately 11% and 1% of the atoms are excited to the first and the second transversely excited states in 4.8 s for $N = 130$. These fractions are larger than the ones introduced by heating, meaning that energy is transferred from longitudinal to transverse directions in the cradle experiment.

In Fig. 9, we compare the experimental data with the MD simulations and observe very good agreement also in the transverse degrees of freedom. The minimal discrepancy from simulations is expected to be caused mainly by the additional excitation during the lattice unloading procedure for band-mapping. This conjecture is demonstrated by measuring $\eta_1$ in the first half oscillation period after the Bragg pulses (shown in the inset of Fig. 9). The additional excitation is clearly observed when the atoms are at the oscillation phase with large momenta at the end of the unloading procedure. For this reason, we accept the local minimum of $\eta_1$ in a half oscillation period as the most credible measurement of the in-trap transverse excitation; for example, the result at $t = 3$ ms represents the situation in the first half period. This effect becomes inevitable as the oscillations of the particle are diffused and becomes weak at longer times as most of the atoms are scattered to the low-energy regime.

## 8 Starting with stronger excitations of the longitudinal motion

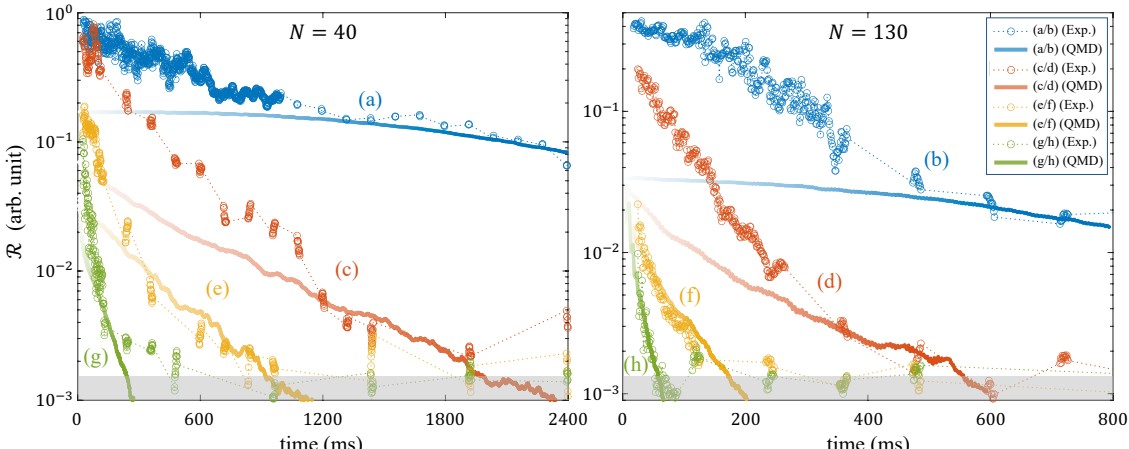

Figure 10: Evolution of $\mathcal{R}(t)$ for the dynamics initialized by the distributions shown in Fig. 3. The results are labeled consistently with Fig. 3. The dynamics of quasimomentum distribution (QMD) calculated in MD simulations (solid curves) are in good agreement with the experimental measurements (open circles) in Stage II of each evolution. The faded-out parts of MD results deviate from the experimental measurements due to the discrepancy between MDF and quasimomentum distribution in the regime far from the non-degenerate limit. The shaded areas indicate the noise floor.

In this section, we go deeper into the crossover regime by initializing the atoms in the 1D tubes with considerably higher longitudinal momenta (energies) (Fig. 3(c)-(h)). Atoms that are kicked to large momenta $\pm 4k_{Bragg}$ and $\pm 6k_{Bragg}$ obtain enough energy to be transversely excited through subsequent collisions. Compared to the dynamics discussed in Sec. 3-7,

these high energy collisions rapidly drive the system out of 1D integrability and towards thermalization. In Fig. 10, we present the evolution of $\mathcal{R}(t)$ for the dynamics initialized by the distributions shown in Fig. 3. For comparison, the branches (a) and (b) are replicas of the results shown in Fig. 7. The dynamics starting with higher imprinted energies exhibit faster relaxation.

The data is again compared to the MD simulations implemented for describing the dynamics. The initial quasimomentum distribution for the simulation is obtained by assuming the same respective particle number in each momentum peak with the experimentally measured initial MDF. The discrepancy between the MDF and the quasimomentum distribution reduces faster in the case with higher energies. Apart from the density reduction due to the single-body dephasing, the increase of temperature also drives the system out of the degenerate regime, especially in the high-energy cases. The MD simulations show good agreement with experimental observations in Stage II (after the quasimomentum distribution coincides with the MDF). The time scale for reaching an MDF that is indistinguishable from a Gaussian (the time when $\mathcal{R}(t)$ drops below the noise floor) is accurately predicted by the simulation. The validity of the semi-classical model is confirmed throughout the regimes from 1D to the dimensional crossover.

# 9 Integrability breaking through virtual excitations

Up to now, we considered only collisions with enough energy to excite the transverse degrees of freedom. However, even when the two-body collision energy is below the threshold of populating the transverse states, the latter can be virtually excited. A collision with a third atom can return the system on the energy shell and simultaneously redistribute momenta of the three-atoms involved in such an effective three-body collision [18, 21]. However, this integrability-breaking mechanism does not contribute much to the relaxation of MDF in our experiment. Indeed, the rate of the velocity-changing collisions per atom due to this mechanism is given by [22]

$$\Gamma_\infty = \frac{2[18\ln(4/3)]^2}{3\sqrt{3}} \frac{\hbar n_{1D}^2 g_2(0)}{m} \left(\frac{a_s}{l_\perp}\right)^4. \tag{3}$$

Here, $l_\perp = \sqrt{\hbar/m\omega_\perp}$, $a_s$ is the 3D $s$-wave scattering length, and $g_2(0)$ is the density-density correlation function at zero distance. $g_2(0)$ is equal to 2 in a non-degenerate gas, to 1 in a weakly interacting quasicondensate and rapidly tends to zero if $\gamma \to \infty$ [70]. Since the 1D density substantially decreases after the rapid dephasing stage, the typical time $1/\Gamma_\infty$ of the MDF relaxation due to virtual transverse excitation extends well beyond the time scale of our experiment.

# 10 Conclusion

We have investigated the relaxation processes of bosons at the onset of the dimensional crossover from 1D to 3D. We demonstrate that the system relaxes in two stages under different mechanisms. At short times, the single-body dephasing rapidly drives the 1D gases into the non-degenerate regime, during which the momentum distribution function deforms and asymptotically approaches the quasimomentum distribution of Lieb-Liniger model. At longer times, a tiny fraction of atoms in the transversely excited states triggers the transition from 1D to 1D-3D crossover. Subsequently, the system relaxes towards an equilibrium with a Gaussian momentum distribution through inelastic two-body collisions at an accelerating rate. A molecular

dynamics simulation was implemented for efficiently modeling the non-equilibrium dynamics. Meanwhile, we proposed a simple method of estimating the momentum distribution functions in the whole regimes of quantum degeneracy. The numerical results quantitatively fit the experimental observations from short to long time scales in all three dimensions.

Moreover, the long-term dynamics of a Newton's cradle with minimal heating and loss as can be obtained in a red-detuned lattice offers a model system to test theoretical methods for describing the complex dynamics in many-body systems at the point of breaking integrability. Future prospects include detailed studies of integral dynamics and its breakdown in the framework of the recently developed generalized hydrodynamics (GHD) [39,40]. In a first step, we have recently extended the applicability of GHD to the dimensional crossover regime [44] and tested it with the data at short to intermediate time scales. But still, many open questions remain, such as for example the many-body dephasing induced by non-trivial interactions [27], or the effect of atom losses [71]. We hope our investigations presented here will pave the way towards a more comprehensive understanding of the non-equilibrium quantum physics in the dimensional crossover regime and the influence of the effectively compactified dimensions.

# Acknowledgements

We thank Benjamin Lev, Marcos Rigol, David Weiss, Camille Lévêque and Qi Liang for enlightening discussions on the cradle experiments. We are grateful to Hepeng Yao for the help on evaluating the temperatures of 1D gases with quantum Monte Carlo calculations. We appreciate Jean-Sébastien Caux, Alvise Bastianello, and Vincenzo Alba for fruitful discussions on GHD. We also thank Andrew Kanagin for proofreading the manuscript.

**Funding information**    X.C. acknowledges the support by the National Natural Science Foundation of China (Grant No. 11920101004, 91736208). J.S. acknowledges the support by the Austrian Science Fund (FWF) via the SFB 1225 ISOQUANT (I 3010-N27). I.M. and H.-P. S. acknowledge the support by the Wiener Wissenschafts- und Technologiefonds (WWTF) via Grant No. MA16-066 (SEQUEX) and by the Austrian Science Fund (FWF) via Grant SFB F65 (Complexity in PDE systems). X.Z. acknowledges the support by the National Key Research and Development Program of China (Grant No. 2016YFA0301501). F.M. acknowledges the support by the Doctoral Program CoQuS.

# A    Experimental details and data analysis

## A.1    Preparation of 1D gases

The $^{87}$Rb BEC is produced in the Zeeman sublevel $F = 1$, $m_F = -1$ by evaporative cooling in a crossed optical dipole trap. In the final stage of the evaporative cooling, the atomic cloud is levitated by switching on a magnetic field gradient in the vertical direction and decompressed by reducing the trap laser power. The total atom number, $N_{tot}$, is tuned between $1 \times 10^4$ and $1 \times 10^5$ by holding the BEC for different times in a shallow trap, where the BEC is overcooled due to the low trap depth. Afterwards, the BEC cloud is adiabatically transferred from the optical dipole trap to a 2D square optical lattice located in the horizontal plane. To avoid interference, the two lattice beams derived from a fiber laser are detuned 220 MHz from each other and have orthogonal polarization. The beam waist ($w^{\text{ol}} = 145 \, \mu$m) of the optical trapping beam is much larger than the BEC.

During the lattice loading procedure, the lattice depth is exponentially ramped to the max-

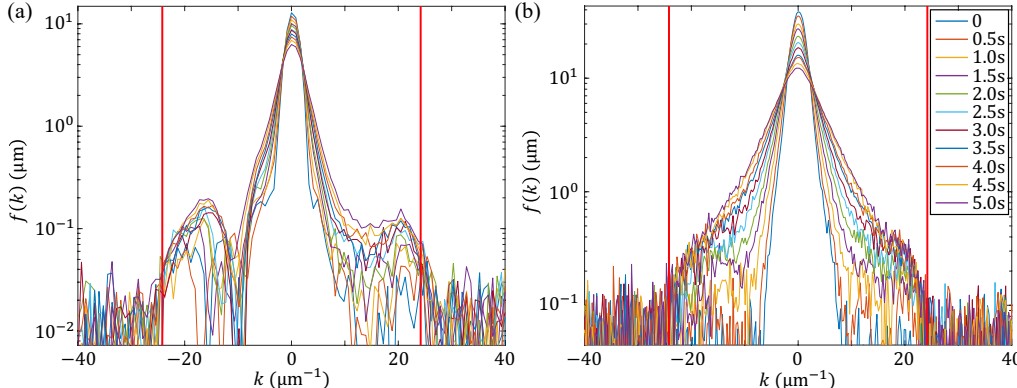

Figure 11: Time evolution of MDF for 1D gases held in lattice without Bragg pulses. (a) $N = 40$; (b) $N = 130$. The broadening of the MDF stems from the heating effect, which is stronger in the case with higher 1D density. The vertical red lines indicate the momenta $\pm k^{\text{th}}$.

imum value $70\,E_R^{\text{ol}}$ in 250 ms (recoil energy $E_R^{\text{ol}} = (\hbar k_R^{\text{ol}})^2/2m$ with the wave vector of optical lattice $k_R^{\text{ol}} = 2\pi/1064$ nm). The optical dipole trap is turned off simultaneously. The atoms are confined by the red-detuned lattice laser both in the vertical (longitudinal) direction with a trap frequency of 83.3(8) Hz, and in the horizontal (transverse) direction with a trap frequency of 31.0(3) kHz. The atoms in different tubes can be regarded as independent 1D gases.

## A.2   Heating process and atomic loss

The heating in an optical lattice is mainly caused by two reasons (*i*) the spontaneous scattering of lattice laser photons; (*ii*) the trap fluctuations (including the laser intensity fluctuations and the pointing stabilities of lattice laser beams) at specific frequencies. The former mechanism heats an atomic system by transferring the photon recoil momenta to atoms. The latter excites atoms to higher transverse states, and the energy may be deposited into the longitudinal kinetic energy through the subsequent inelastic collisions. The heating effect is in general stronger for 1D gases with higher atomic densities because of the larger collision rates.

In our experiments, the heating process is studied by observing the evolution of 1D gases held in the identical lattice setup without the Bragg-pulse excitation. The MDFs for both $N = 40$ and $N = 130$ exhibit the expansion of the momentum peaks (see Fig. 11). By summing up the contribution on each pixel of the MDF measurement, we obtain the increase of kinetic energy of $0.06\hbar\omega_\perp/s$ and $0.09\hbar\omega_\perp/s$ for $N = 40$ and $N = 130$, respectively. On the other hand, we also estimate the heating rate by evaluating the energy growth in the transverse dimensions and observe a rate $0.006\hbar\omega_\perp/s$ for $N = 130$ (see Fig. 12). Most of the transversely excited atoms appear in the first state, while the signal in the second excited state is so weak that it is submerged in the imaging noise.

Although the heating rate in the ground state is usually higher in a red-detuned lattice [28–31], we suppress the heating in our system to a minimal value by the large detuning of the lattice laser and the carefully controlled environment. The heating rates achieved in our experiments are at least twice as low as observed in Ref. [6] in a blue-detuned lattice.

The atom loss observed in our experiments is between $\sim$ 4%/s ($N = 40$) and $\sim$ 7%/s ($N = 130$), which are about one order of magnitude lower than observed in Ref. [6]. Such low loss rates enable us to study the long-term dynamics of bosons out of equilibrium. The loss rates are nearly constant throughout the evolution. We do not observe any significant three-body loss as seen in Ref. [6, 32].

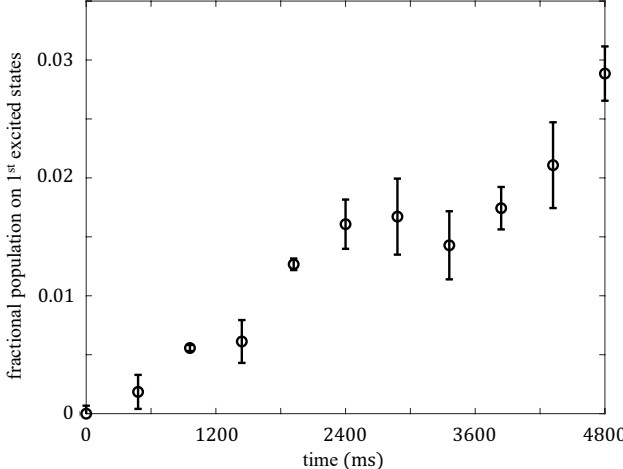

Figure 12: Time evolution of the fraction of the first transversely excited state for 1D gases held in lattice without Bragg pulses ($N = 130$). The error bars denote the standard deviation of five measurements.

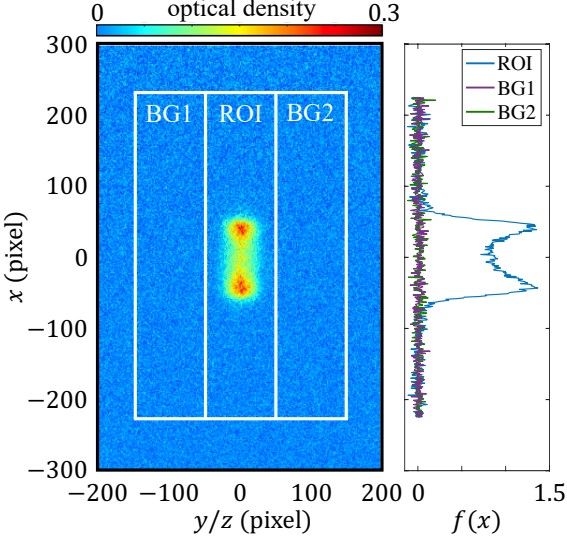

Figure 13: An example of the horizontal imaging. A region of interest (ROI) is arranged to contain all atoms, and two background regions BG1 and BG2 are selected next to ROI. The pixel size is $6.45\mu$m$\times 6.45\mu$m in the object plane. To obtain the longitudinal MDF and estimate the noise floor, we integrate the images in these regions over the transverse direction, respectively. The MDF (blue) and noise level (green and purple) are shown on the right.

## A.3 Detection of longitudinal momentum distribution function

The 1D gases are detected by standard absorption imaging after being released from the lattice trap and expanding in 3D. To keep the signal-to-noise ratio of images at a comparable level, the expansion time for $N = 40$ and 130 are set to 10 ms and 30 ms, respectively. In the horizontal plane, the image is taken along the bisector of two lattice beams. The lattice is turned off in 500 $\mu s$, during which the interparticle interaction vanishes. It is fast compared to the longitudinal dynamics but slow enough to prevent the atomic cloud from spreading too much in transverse directions. By integrating the image over the transverse direction in the region of interest (ROI), we obtain the longitudinal distribution profile. This profile approaches MDF

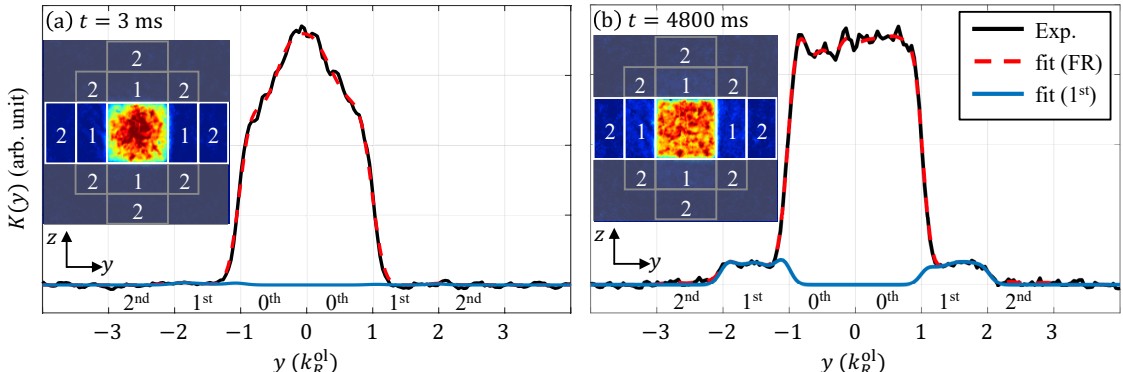

Figure 14: Evaluating the fractions of atoms in transverse states. The method is illustrated with two examples of the band-mapping images taken at (a) the beginning and (b) the end of dynamical evolution ($N = 130$). By integrating the band-mapped distribution in the unshaded region, we obtain the 1D distributions (black solid curves) and fit them to a summation of a set of Gaussian curves in the full range (FR) (red dashed curves). The fitted distribution of the first excited state is shown in blue, separated from the ground state and the second excited states.

$f(k)$ after a long TOF. To assess the impact of the imaging noise, which mainly stems from the photon and atom shot noise on the CCD camera, we choose two background regions (BG1 and BG2) beside the ROI with the same size (see Fig. 13) and consider them as the noise floor in the data analysis.

## A.4   Detection of populations in transverse states

In the vertical direction, band mapping is applied to obtain information of the population in the respective energy-band. The lattice depth is exponentially turned off in 2 ms. During the turnoff, the crystal momentum is mapped to the free particle momentum, and afterwards, the Brillouin zones are imaged.

Two examples are shown in Fig. 14 to explain the method of evaluating the fractional population of atoms in each state. White boxes separate the 2D band-mapped distributions into Brillouin zones corresponding to the ground state, the first and second excited states. The biggest issue to overcome for achieving an accurate fraction in each state is the overlap between adjacent Brillouin zones due to the broadening of the quasi-free-particle momentum distribution. Firstly, we integrate the band-mapped distribution in the central region (the unshaded area) in $z$ direction over the width of the first Brillouin zone. Secondly, we fit the integrated distribution $K(y)$ to a summation of 180 Gaussian curves arranged with equal spacing and identical r.m.s. width between $y = -3\hbar k_R^{ol}$ and $y = +3\hbar k_R^{ol}$

$$K(y) = \sum_{y_c = -3\hbar k_R^{ol}}^{+3\hbar k_R^{ol}} A(y_c) \exp\left[-\frac{(y - y_c)^2}{2\sigma^2}\right], \tag{4}$$

where $A(y_c)$ is the amplitude of the Gaussian curve centered on $y_c$ [4]. By integrating $A(y_c)$ in the corresponding regions, we get the fractional population of atoms in each state. The same calculation is repeated in the $z$ direction, and $\eta_1$, $\eta_2$ are calculated according to the results from both $y$ and $z$ dimensions. Since the atoms in the second excited state located in the four

---

[4]In comparison with the method of deconvolution, our processing approach is free of the ill-posed problem.

corners are not included in the calculations in both dimensions, $\eta_2$ is multiplied by 1.5 under the assumption of a uniform distribution.

## B  Model of the atomic collision in molecular dynamics calculation

In this section, we describe how atomic collisions are modeled. Since the system is 1D and quasiparticles are indistinguishable, we can always consider an ordered array of quasiparticles, $x_1 < x_2 < \ldots < x_{N-1} < x_N$. In what follows, it is convenient to introduce the scaled coordinates $\bar{x}_j = x_j / l_\parallel$ and quasimomenta $\bar{q}_j = q_j l_\parallel$, where $l_\parallel = \sqrt{\hbar/(m\omega_\parallel)}$.

For a given configuration of $N$ quasiparticles in the phase space we calculate the time of the first collision, i.e. the first (smallest) time when the coordinate of any two neighboring quasiparticles coincide. The oscillatory motion of the $j$th atom is described by

$$\begin{aligned}
\bar{x}_j(t+\tau) &= \bar{x}_j(t)\cos\omega_\parallel\tau + \bar{q}_j(t)\sin\omega_\parallel\tau, \\
\bar{q}_j(t+\tau) &= -\bar{x}_j(t)\sin\omega_\parallel\tau + \bar{q}_j(t)\cos\omega_\parallel\tau.
\end{aligned} \tag{5}$$

Then we calculate the collision time $\tau_j$ for the $j$th and $(j+1)$th quasiparticles:

$$\tan\tau_j = -\omega_\parallel^{-1}\frac{\bar{x}_j(t) - \bar{x}_{j+1}(t)}{\bar{q}_j(t) - \bar{q}_{j+1}(t)}, \qquad \tau_j > 0,$$

and find the smallest one,

$$\tau = \min_{1\le j\le N-1}\tau_j.$$

We propagate the quasiparticles until the time $t+\tau$ according to Eq. (5) and then decide, according to the probabilities (see below) and using a pseudorandom number generator, what happens to the transverse states of the involved quasiparticles. The probabilities of the change of the transverse state are based on the standard quantum mechanical expressions, which can be easily derived for a pair of colliding quasiparticles with the initial state of their relative motion in the $(y, z)$-plane as the transverse ground state [46]. However, for the sake of simplicity, we neglect any dependence of the transverse transition probabilities on the transverse quantum states of colliding quasiparticles. As such, the scheme of transverse transitions is simplified.

The collisions are assumed to be instantaneous, in other words, the Wigner delay time is neglected. This is justified by the observation that, even though the interplay between Wigner delay time and the longitudinal trapping potential will lead to appreciable thermalization, in our non-degenerate system this time scale is very long [23] and exceeds the duration of the experiment.

To be definite, consider a collision of the quasiparticles 1 and 2. Their coordinates at the collision time are $x_1 = x_2$ and the respective quasimomenta are $\hbar q_1$ and $\hbar q_2$. The quasimomentum of the relative motion is canonically conjugate to the interatomic distance $x_2 - x_1$ and defined as

$$\hbar q = \frac{1}{2}\hbar(q_2 - q_1).$$

The total quasimomentum of the pair is denoted by

$$\hbar Q = \hbar(q_1 + q_2).$$

Concerning the transverse quantum numbers, we begin with the option

$$n_1 = n_2.$$

Because of the parity conservation, the transverse energy of a pair of quasiparticles in the course of a collision can change by a multiple of $2\hbar\omega_\perp$. If the kinetic energy of the relative motion is less than $2\hbar\omega_\perp$, then the increase of the transverse energy is impossible. In the opposite case,

$$\frac{\hbar^2\tilde{q}^2}{m} = \frac{\hbar^2 q^2}{m} - 2\hbar\omega_\perp > 0,$$

the increase of the transverse energy by $2\hbar\omega_\perp$ is possible. The probability of such an event is

$$\mathcal{P}_\uparrow = \frac{2q\tilde{q}\alpha_{1D}^2}{q^2\tilde{q}^2 + \alpha_{1D}^2(q+\tilde{q})^2}, \tag{6}$$

where $\alpha_{1D} = c/2$. A pseudorandom number $\zeta$ uniformly distributed between 0 and 1 is generated. If $\zeta < \mathcal{P}_\uparrow$ then we raise the transverse excitation energy by 2 quanta. To preserve the ordering of quasiparticles in the course of the subsequent evolution, we assign the new (primed) quasimomenta to them as follows:

$$\hbar q_1' = \hbar\left(\frac{1}{2}Q - \tilde{q}\right), \qquad \hbar q_2' = \hbar\left(\frac{1}{2}Q + \tilde{q}\right).$$

With the help of a new pseudorandom number we assign the new transverse quantum numbers with the following probabilities:

$$
\begin{array}{llr}
n_1' = n_1, & n_2' = n_1 + 2 & (25\,\%), \\
n_1' = n_1 + 2, & n_2' = n_1 & (25\,\%), \\
n_1' = n_1 + 1, & n_2' = n_1 + 1 & (50\,\%).
\end{array}
$$

The detailed balance condition should be satisfied: the number of transitions up and down per unit time must be the same on average. Therefore, if

$$n_1 = n_2 > 0$$

then we allow for the transition to the state characterized by

$$n_1' = n_1 - 1, \qquad n_2' = n_1 - 1,$$

and

$$\hbar q_1' = \hbar\left(\frac{1}{2}Q - \tilde{Q}\right), \hbar q_2' = \hbar\left(\frac{1}{2}Q + \tilde{Q}\right), \tag{7}$$

where

$$\frac{\hbar^2\tilde{Q}^2}{m} = \frac{\hbar^2 q^2}{m} + 2\hbar\omega_\perp.$$

The probability of this process is

$$\mathcal{P}_{n_1,n_1 \to n_1-1,n_1-1} = \left(\frac{n_1}{n_1+1}\right)^2 \mathcal{P}_\downarrow, \tag{8}$$

where

$$\mathcal{P}_\downarrow = \frac{2q\tilde{Q}\alpha_{1D}^2}{q^2\tilde{Q}^2 + \alpha_{1D}^2(q+\tilde{Q})^2}. \tag{9}$$

The prefactor in front of $\mathcal{P}_\downarrow$ in Eq. (8) ensures the detailed balance. The condition of the downward transverse transition corresponds to the pseudorandom number $\zeta$ falling between $\mathcal{P}_\uparrow$ and $\mathcal{P}_\uparrow + \mathcal{P}_{n_1,n_1 \to n_1-1,n_1-1}$.

If, finally, $\zeta > \mathcal{P}_\uparrow + \mathcal{P}_{n_1, n_1 \to n_1-1, n_1-1}$ then no change of the transverse states takes place. To maintain the ordering of atoms in this case, we set

$$\hbar q_1' = \hbar q_2, \qquad \hbar q_2' = \hbar q_1.$$

This is always the case when two quasiparticles in the ground transverse states collide with the energy insufficient for excitation by two transverse quanta.

Consider now another possibility

$$n_1 \neq n_2.$$

Here an important simplification of the model comes into play. If the transverse states of colliding quasiparticles are different, we neglect, except of a special case described below, the change of the set of the transverse excitation numbers, allowing for the exchange process only, when the transverse excitation numbers associated with the two quasimomenta $\hbar q_1$ and $\hbar q_2$ are interchanged:

$$\begin{aligned}
\hbar q_1' &= \hbar q_2, & \hbar q_2' &= \hbar q_1, \\
n_1' &= n_1 & n_2' &= n_2.
\end{aligned}$$

The probability of the exchange process is given by

$$\mathcal{P}_{\text{ex}} = \frac{1}{2} \frac{\alpha_{1\text{D}}^2}{q^2 + \alpha_{1\text{D}}^2}. \tag{10}$$

A special case is given by

$$n_2 = n_1 + 2 \qquad \text{or} \qquad n_1 = n_2 + 2.$$

In this case, in addition to the exchange process, the decrease of the larger of the transverse excitation numbers by 2 can happen, as it is required by the detailed balance:

$$n_1' = n_2' = \min(n_1, n_2)$$

and the quasimomenta after collision are given by Eq. (7). The respective probability is given by

$$\mathcal{P}_{|n_1-n_2|=2 \to n_1=n_2} = \frac{1}{2} \frac{\min(n_1, n_2) + 1}{\min(n_1, n_2) + 3} \mathcal{P}_\downarrow, \tag{11}$$

where $\mathcal{P}_\downarrow$ is again given by Eq. (9).

## C Estimation of bosonic momentum distribution function

For a degenerate 1D Bose gas, the MDF, defined as the Fourier transform of the correlation function, is distinct from the quasimomentum distribution within the Lieb-Liniger model. In the regime where the temperature is low (below the chemical potential) and the Lieb-Liniger parameter $\gamma$ is not excessively large, the correlation function for bosonic Luttinger liquid at $x \gg \hbar/mc_s$ is written as

$$g_1(x) = \frac{1}{n_{1\text{D}}} \langle \hat{\Psi}^\dagger(x) \hat{\Psi}(0) \rangle = C_0 \left[ \frac{1}{\sinh(\pi k_T |x|)} \right]^{\frac{1}{2K}}, \tag{12}$$

where $C_0 \sim 1$, $c_s$ is the speed of sound, $k_T = k_B T/(\hbar c_s)$ and $K = \pi \hbar n_{1\text{D}}/(mc_s)$ is the Luttinger liquid parameter. The MDF $W(k) = \int_{-\infty}^{+\infty} \frac{dx}{2\pi} g_1(x) e^{ikx}$, and it is expressed via Euler's beta-function [60]

$$W(k) = \frac{C_0 2^{\frac{1}{2K}}}{2\pi^2 k_T} \text{Re} \left[ B \left( \frac{ik}{2\pi k_T} + \frac{1}{4K}, \, 1 - \frac{1}{2K} \right) \right], \tag{13}$$

where $B(x, y) = \Gamma(x)\Gamma(y)/\Gamma(x+y)$. $W(k)$ consists of a Lorentzian profile on top of a pedestal. The central Lorentzian is restricted to $k < k_T$, and $k_T$ is the momentum where the Bose-Einstein distribution starts to deviate from the Rayleigh-Jeans classical limit and to decrease exponentially. For larger momenta $k \gg k_T$, $W(k)$ decreases $\propto \text{const}/k$, slower than Lorentzian.

For much larger momenta $k \gg k_C$, $W(k)$ is determined by Tan's contact and decreases $\propto Ck^{-4}$, where $C$ is the Tan's contact and the momentum $k_C$ is approximately equal to the maximum quasimomentum at zero temperature. There are known approaches to precisely calculate the value of Tan's contact, see, for example, Ref. [51]. Considering the experimental uncertainties, we use an asymptotic approximation for large momenta $k \gg \xi_h^{-1}$, where $\xi_h$ is the healing length, for a weakly interacting quasicondensate. For stronger interactions, the qualitative picture is similar. The modified MDF is

$$\widetilde{W}(k) = \frac{W(k)}{\sqrt{1 + \frac{1}{4}(k\xi_h)^2 \left[1 + \frac{1}{2}(k\xi_h)^2 + k\xi_h\sqrt{1 + \frac{1}{4}(k\xi_h)^2}\right]}}. \tag{14}$$

For $k \gg \xi_h^{-1}$, $\widetilde{W}(k) \propto k^{-4}$. In general, the $\widetilde{W}(k)$ expressed by Eq. (13) and (14) is expected to be much narrower than the corresponding quasimomentum distribution.

In the non-degenerate limit, the MDF coincides with the quasimomentum distribution. Furthermore, when the temperature $T$ is relatively high, the MDF approaches the Maxwell-Boltzman distribution,

$$M(k) = \frac{1}{\sqrt{2\pi m k_B T/\hbar^2}} \exp\left(-\frac{\hbar^2 k^2}{2m k_B T}\right). \tag{15}$$

Let us consider that we try to derive the MDF corresponding to a distribution of quasiparticles $\rho^{\text{target}}(x, q)$, namely the target distribution. The basic idea to estimate the MDF is done by fitting the target distribution with a sum of multiple thermal distributions $\rho(x, q) = \sum_i \rho_i(x, q)$, which are calculated by solving the Bethe-ansatz equations in a harmonic trap defined with experimentally measured parameters. In a general case of quantum Newton's cradle experiments, the $\rho(x, q)$ consists of three components of quasiparticles, described by thermal distributions $\rho_1$, $\rho_2$ and $\rho_3$, respectively. The central component $\rho_1$ centers at the origin of the phase space with zero mean quasimomentum ($\langle q_1 \rangle = 0$). The symmetric Bragg components $\rho_2$ and $\rho_3$ are derived from the Bragg pulses, and they are shifted by the mean quasimomentum boosts $\langle q_2 \rangle$ and $\langle q_3 \rangle$ ($\langle q_2 \rangle = -\langle q_3 \rangle = 2k_{Bragg}$). All of $\rho_i$ are normalized to the respective quasiparticle number $N_i$, subject to the restriction $\sum_i N_i = N$.

To imitate the effect of single-body dephasing, we drop the densities of $\rho_2$ and $\rho_3$ by a scale factor $\alpha_n$, which is calculated by assuming a linear expansion of the Bragg components in the direction of motion. $\alpha_n$ decreases from 1 at $t = 0$ and stops decreasing when two Bragg components merge until fully dephased. The expanding rate is determined by evaluating $\mathcal{D}(t)$ for the calculated distributions and comparing the results with experimental observations as shown in Fig 6.

To make the fitting procedure simple and fast, we integrate the distributions $\rho^{\text{target}}(x, q)$ and $\rho(x, q)$ over the direction of motion and seek for the minimum discrepancy between the distributions in the radial coordinate. We accept this simplification in a quasi-harmonic potential when the interaction is not excessively large. The distribution in the radial coordinate hardly changes within one period of oscillation. The best-fit distribution returns us the chemical potential $\mu_i$ and temperature $T_i$ for each component. Following Eq. (13-15), the MDFs in the degenerate and non-degenerate limits are derived.

As we discussed in the main text, the cradle system evolves from the degenerate regime to the non-degenerate regime during the dynamical evolution. To interpolate the crossover

between two limits, we propose an empirical formula: convolving the MDF for the degenerate limit $\widetilde{W}_i(k)$ with its counterpart for the non-degenerate limit $\widetilde{M}_i(k)$

$$f_i(k) = \int dk' \widetilde{W}_i(k-k')\widetilde{M}_i(k').$$ (16)

Here we modify Eq. (15) via $\widetilde{M}_i(k) = M_i(k/\beta)$ so that we rescale the width of the profile. $\beta$ is tuned from 0 to 1 in the crossover regime and it follows $\beta \propto (\mu_i/T_i)^2$ when $\mu_2 < 0$. For the central component $\rho_1$, $\mu_1 \ll -T_1$ is obtained throughout the entire evolution of time, resulting in an MDF very close to its corresponding quasimomentum distribution. While for the Bragg component $\rho_2$ and $\rho_3$, $\mu_2$ and $\mu_3$ evolve from positive to negative and in longer times becomes much smaller than $-T_2$ and $-T_3$. Thus, we obtain peaked MDFs in the early stage of evolution and gradually rounded MDFs that asymptotically approach the quasimomentum distributions as the system evolves towards the non-degenerate limit.

Since the mean quasimomentum equals to the mean momentum, $f_i(k)$ is shifted to be centered at $\langle q_i \rangle$. The MDF for the entire cloud $f(k) = \sum_i f_i(k)$.

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
