# Peer review of "Relaxation of Bosons in One Dimension and the Onset of Dimensional Crossover"

_SciPost Physics, doi:SciPost Phys. 9, 058 (2020)_

## Round 2 · Referee Report · Anonymous (Referee 1) · 2020-9-11

Report

In this manuscript the authors study experimentally the breaking of integrability for a 1D Bose gas, at the onset of the crossover from one to three spatial dimensions. The experimental setting is reminiscent of the famous quantum Newton’s cradle (Ref.[6]); however, within the suggested set up the observation time is extended significantly beyond Ref. [6], allowing for a study of both the longitudinal momentum distribution function and the transverse excitations.

The main result reported in this work is the observation of a two-stage thermalization process: a first stage characterized by the single-body dephasing, where integrability is nearly preserved, and a second stage, where inelastic collisions involve the transverse excitation modes, leading to the dimensional crossover.

The experimental data are explained in terms of a combination of integrability-based methods. Crucially, the presence of stable quasi-particles is at the root of the success of the molecular-dynamic simulations employed, which, in turn, are known to provide a very good approximation to the generalized hydrodynamic predictions.

I find the draft extremely well written. Furthermore, the topic is certainly very timely, and of great interest for a broad audience, given the widespread current interest in the hydrodynamic description of integrable systems. In addition, the authors manage to present their results in a very clear way, despite the fact that the theoretical predictions necessarily involve highly technical numerical calculations and analytical background.

I have only one minor comment, related to a terminology used by the authors. Especially in the introduction, it is often repeated that integrable systems do not "relax", but "dephase" towards a GGE. This terminology might appear confusing for a theoretical audience. Usually, the term relaxation is used as a synonym of "local equilibration", and indeed integrable systems do locally equilibrate to GGEs. The authors, instead, seem to use the term relaxation with the meaning of thermalization. They might consider making use of the latter term, to eliminate any possible source of confusion.

In conclusion, for the above reasons, I highly recommend publication in Scipost Physics.
  • validity: -
  • significance: -
  • originality: -
  • clarity: -
  • formatting: -
  • grammar: -

Author:  Chen Li  on 2020-10-06  [id 994]

(in reply to Report 1 on 2020-09-11)
Category:
answer to question

We thank the referee for her/his positive reports and constructive suggestions. Especially we thank the referee for clarifying the terminology and the use of relaxation/thermalization and dephasing for the case of integrable systems. We have revised the relevant sentences in paragraphs 1 and 2 in the introduction.

In our Newton’s cradle experiments, the integrability is broken and quasiparticles interact with each other (weakly). In this situation, by dephasing we mean the dephasing of quasiparticles on time scales when their interaction can be neglected, and by relaxation we mean the more complicated dynamics as the result of the interaction of quasiparticles. Our notion of relaxation is also related to local equilibration in the momentum space. The MDF evolves towards equilibrium, which may be viewed as local equilibration dynamics of individual momentum states, the other ones representing a bath.

For us as experimentalists, relaxation is much more general than thermalization which characterizes the evolution towards a thermal equilibrium state. Thermalization is also much harder to prove than relaxation. In our experiment, we look mainly at one observable, the momentum distribution, and observe relaxation towards a Gaussian distribution. This is only an indication of thermalization but no proof. For that we would have to show that the final state we observe is not some long-lived pre-thermal state which is not fully thermalized in all its components. Therefore, from an experimental point of view, we feel much more comfortable to call the observed time evolution in the momentum distribution relaxation.

---

## Round 2 · Referee Report · Anonymous (Referee 2) · 2020-9-18

Strengths

  1. Compelling comparison between experimental and theoretical results

  2. Experiment realized on a very long time scale (several hundreds of trap periods)

  3. Fundamental results on a timely topic: new insights on the mechanisms for dephasing and thermalization in the quantum Newton cradle setup

  4. Overall the presentation is clear and well organized

Weaknesses

  1. Minor weakness: some details could be clarified (see below)

Report

The authors present a detailed analysis of the mechanisms for thermalisation in the quantum Newton's cradle setup. Thanks to an elaborate mixture of state-of-the-art experimental techniques and advanced theoretical modeling, they identify a two-stage process which brings the one-dimensional Bose gas to equilibrium. During the first few oscillation cycles, the approach to a slowly evolving quasi-stationary state is caused by single-particle dephasing. Then at later time the system keeps evolving slowly towards an equilibrium distribution because of transverse excitations which break integrability.

This paper sheds light on a long-standing fundamental issue in the literature: does the Newton's cradle thermalise and if so, how and on what time scales?
Therefore it satisfies (at least) one of the acceptance criteria of Scipost Physics ("breakthrough on a previously-identified and long-standing research stumbling block") and it clearly deserves to be published there.

A few points should be clarified before publication (see below); I also noticed some typos and I encourage the authors to proofread their manuscript again.

Requested changes

  1. the authors should clarify what they mean by 'molecular dynamics' ('MD') simulations. In the present version it is rather confusing. When this terminology is first introduced in page 3, the authors immediately cite Refs. [35-37]. But Refs. [36,37] are references on GHD, therefore the 'molecular dynamics' there includes a Wigner time delay at each two-body collision. On the contrary, the MD simulations performed by the authors do not seem to include this Wigner time delay (so they are truly different from a solution to the GHD equations). The precise meaning of 'MD' should be explained, as it is an important ingredient of the paper. This is especially unclear in Sec. 6 where the authors describe their MD simulation in a way that really sounds like GHD: 'we treat the atoms as quasiparticles characterized by their spatial coordinates and quasimomenta' (yet the authors seem to make a distinction between GHD and MD).

  2. throughout the paper, the authors refer to the 'degenerate' and 'non-degenerate' regimes, without always giving enough information to convince the reader that the gas is indeed in the claimed regime. For instance, in Sec. 6: the theory there is based on the assumption that the gas is in the non-degenerate regime so that the atoms behave like non-interacting bosons. But no justification of this assumption seems to be given. (Perhaps the authors could give $\gamma$ and $t$ so that one can check that $\gamma \ll t^{-1/2}$; or give another justification.)

  3. what exactly is plotted in Fig. 3, is that experimental data or numerics (e.g. Monte Carlo)? (If it is numerics, how is it obtained exactly?)

  4. In Sec. 2, when the authors present the experimental setup, they say that the longitudinal frequency is $\omega_\parallel = 2\pi \times 83.3 {\rm Hz}$, as if that frequency was the same for all the tubes. But isn't there some inhomogeneity between the tubes? If so it should be made clear in the description of the experimental setup. (The same question holds for the transverse frequency $\omega_\perp$.)

  5. A more minor point: in the introduction, the authors say 'hard-core' as if it were a synonym of 'short range'; but doesn't 'hard-core' usually imply infinite repulsion?

  6. In the caption of Fig. 4, have the authors checked whether the different oscillations periods are consistent with the anisotropy of the trap?

  7. In a few places the authors use the name 'quasimomentum' while in other places they use 'rapidity'. I assume the two are the same; it would be clearer if a single term was used consistently throughout the paper.

  8. Finally I noticed typos, for instance: -in page 4, 'independent with the 1D density' -page 14, 'in contract to' instead of 'in contrast to'

  • validity: top
  • significance: top
  • originality: high
  • clarity: high
  • formatting: excellent
  • grammar: good

Author:  Chen Li  on 2020-10-06  [id 996]

(in reply to Report 2 on 2020-09-18)
Category:
answer to question

We thank the referee for the insightful comments and for the constructive suggestions. Below we provide detailed answers to the questions:

Q1. the authors should clarify what they mean by 'molecular dynamics' ('MD') simulations. In the present version it is rather confusing. When this terminology is first introduced in page 3, the authors immediately cite Refs. [35-37]. But Refs. [36,37] are references on GHD, therefore the 'molecular dynamics' there includes a Wigner time delay at each two-body collision. On the contrary, the MD simulations performed by the authors do not seem to include this Wigner time delay (so they are truly different from a solution to the GHD equations). The precise meaning of 'MD' should be explained, as it is an important ingredient of the paper. This is especially unclear in Sec. 6 where the authors describe their MD simulation in a way that really sounds like GHD: 'we treat the atoms as quasiparticles characterized by their spatial coordinates and quasimomenta' (yet the authors seem to make a distinction between GHD and MD).

A1. The molecular dynamics is a numerical approach that calculates the classical (Newtonian) trajectories of atoms or molecules. Their intrinsic degrees of freedom may be discrete (quantized). This is a widely used approach in physics of fluids and in physical chemistry. The standard assumption of MD is that the system is non-degenerate, i.e., the de Broglie wavelength is shorter than the mean interparticle distance. Therefore, we can use Newtonian physics instead of quantum mechanics for center-of-mass motion. Refs [36, 37] (Refs [37, 38] in the revised manuscript) are two examples of how MD is mapped to Generalized Hydro Dynamics (GHD) and how MD is applied to solving GHD problems. The main advantage of the MD in comparison to the continuum models (such as classical hydrodynamics) is that the former approach deals with solving ordinary differential equations (albeit for many particles simultaneously) and sometimes exhibits very good efficiency. In our experimental case, MD can successfully substitute the full GHD just because in the non-degenerate regime collective excitations are reduced to individual atoms and the respective quasimomenta become equal to the momenta, which determine the trajectories of the atoms.
To satisfy a broad audience and eliminate any possible confusion, we added four sentences and one reference (Ref [35]) as a general introduction of MD in Sec. 1 Para. 5. Meanwhile, we reformulated Sec. 6 Para. 2 to describe our MD method clearer.

Q2. throughout the paper, the authors refer to the 'degenerate' and 'non-degenerate' regimes, without always giving enough information to convince the reader that the gas is indeed in the claimed regime. For instance, in Sec. 6: the theory there is based on the assumption that the gas is in the non-degenerate regime so that the atoms behave like non-interacting bosons. But no justification of this assumption seems to be given. (Perhaps the authors could give $\gamma$ and $t$ so that one can check that $\gamma \ll t^{-1/2}$; or give another justification.)

A2. The initial reduced temperature $\widetilde{T}$ (the symbol $t$ for reduced temperature was replaced by $\widetilde{T}$ to distinguish it from time) and the coupling strength $\gamma$ (before and after dephasing) are both given in Sec. 2.1. $\gamma$ increases by a factor of ~4 during the dephasing process, so that the conditions $\gamma \widetilde{T}^{1/2}>7.6 \ (N=40)$ and $\gamma \widetilde{T}^{1/2}>5.2 \ (N=130)$ are fulfilled for dephased distributions. Please note that the coupling strength grows not by itself, but in comparison to the linear density. Here we assume a constant temperature during the dephasing. But in the real case, any increase in temperature resulting from relaxation and heating will make $\gamma \widetilde{T}^{1/2}$ even larger. Furthermore, in the first ten periods of oscillation, the chemical potential turns from positive to negative, suggesting a transition from the quantum correlated dynamics to the dynamics of individual atoms. In the revised manuscript, we added more details in Sec. 5 Para. 2 and Sec. 6 Para. 2 to stress this evidence.

Q3. what exactly is plotted in Fig. 3, is that experimental data or numerics (e.g. Monte Carlo)? (If it is numerics, how is it obtained exactly?)

A3. Fig. 3 shows the experimental measurements, and it is clarified in the caption of Fig. 3 in the revised manuscript.

Q4. In Sec. 2, when the authors present the experimental setup, they say that the longitudinal frequency is $\omega_\Vert=2 \pi \times 83.3$Hz, as if that frequency was the same for all the tubes. But isn't there some inhomogeneity between the tubes? If so it should be made clear in the description of the experimental setup. (The same question holds for the transverse frequency $\omega_\perp$.)

A4. We added the variations of $\omega_\Vert$ and $\omega_\perp$ over the occupied tubes in Sec. 2.1 Para. 1 and Appendix A.1 Para. 2.

Q5. A more minor point: in the introduction, the authors say 'hard-core' as if it were a synonym of 'short range'; but doesn't 'hard-core' usually imply infinite repulsion?

A5. We agree that the term 'hard-core' is inappropriate here. It has been removed in the revised manuscript.

Q6. In the caption of Fig. 4, have the authors checked whether the different oscillations periods are consistent with the anisotropy of the trap?

A6. We have checked that the measurements in Fig.4 are consistent with the expectation based on our trap parameters within reasonable experimental imperfections, for example, the nonideal lattice beam quality, the imperfect overlap between lattice beams, etc. This is declared in Sec. 3 Para. 1 in the revised manuscript.

Q7. In a few places the authors use the name 'quasimomentum' while in other places they use 'rapidity'. I assume the two are the same; it would be clearer if a single term was used consistently throughout the paper.

A7. The 'quasimomentum' and 'rapidity' are indeed the same in our manuscript. We have revised the description to keep it consistent throughout the manuscript.

Q8. Finally I noticed typos, for instance:
-in page 4, 'independent with the 1D density'
-page 14, 'in contract to' instead of 'in contrast to'

A8. We have corrected the typos pointed out by the referee and proofread the manuscript again for any remaining errors.

---

## Round 3 · Author Response

We thank the referees for the constructive suggestions. Their comments are addressed with the changes listed below. And more details are described in our replies to the referees.

---

## Round 3 · List of Changes

• According to Referee 1's comments, in Sec. 1 Para. 1-2, we replaced the term 'relax' with 'thermalize' and revised the relevant statements to eliminate any possible confusion.
  • According to Referee 2's comment 1, in Sec. 1 Para. 5, a general introduction of molecular dynamics (MD) and Ref [35] were added to satisfy a broad audience. Additionally, Sec. 6 Para. 2 was formulated to improve the description of our molecular dynamics method.
  • According to Referee 2's comment 2, in Sec. 5 Para. 2 and Sec. 6 Para. 2, we offered the justification of our statement that the dynamics mostly happened in the non-degenerate regime.
  • According to Referee 2's comment 3, in the caption of Fig.3, it was clarified that the plots showed experimental measurements.
  • According to Referee 2's comment 4, in Sec. 2.1 Para. 1 and Appendix A.1 Para. 2, the variations of ω_∥ and ω_⊥ over the occupied tubes were clarified.
  • According to Referee 2's comment 5, in Sec. 1 Para. 1, an inappropriate statement was removed.
  • According to Referee 2's comment 6, in Sec. 3 Para. 1, we declared that the results shown in Fig. 4 were consistent with the expectations.
  • According to Referee 2's comment 7, we replaced 'rapidity' with 'quasimomentum' so that the terminology is consistent throughout the manuscript.
  • Typos were corrected.
  • The affiliation and funding information of Igor Mazets and Hans-Peter Stimming have been updated.

---

## Editorial Decision

published